# Non-HLA Autoantibodies Against Angiotensin II Receptor 1 (AT1R) and Endothelin A Receptor (ETAR) in Pediatric Kidney Transplantation

**DOI:** 10.3390/ijms252111817

**Published:** 2024-11-03

**Authors:** Benedetta Antoniello, Susanna Negrisolo, Diana Marzenta, Marta Vadori, Piera De Gaspari, Emanuele Cozzi, Elisa Benetti

**Affiliations:** 1Laboratory of Immunopathology and Molecular Biology of the Kidney, Department of Women’s and Children’s Health, University of Padova, 35127 Padua, Italy; benedetta.antoniello@studenti.unipd.it (B.A.); susanna.negrisolo@unipd.it (S.N.); diana.marzenta@aopd.veneto.it (D.M.);; 2Pediatric Research Institute “IRP Città della Speranza”, 35127 Padua, Italy; 3Pediatric Nephrology, Department of Women’s and Children’s Health, Padua University Hospital, 35128 Padua, Italy; 4Transplant Immunology Unit, Department of Cardiothoracic-Vascular Sciences and Public Health, Padua University Hospital, 35128 Padua, Italy; 5Euroimmun Italy, Laboratory Reference, 35127 Padua, Italy

**Keywords:** renal transplant, antibody-mediated rejection, pediatric kidney transplantation, protocol biopsy, autoantibodies, AT1R, ETAR, ICAM-1, VCAM-1

## Abstract

Antibody-mediated rejection (AMR) is the leading cause of premature kidney transplant failure. The role of alloantibodies against Human Leukocyte Antigens (HLA) has been a primary focus in AMR. More recently autoantibodies and alloantibodies against the angiotensin II receptor type 1 (AT1R) and the endothelin A receptor (ETAR) have been linked to poor allograft outcomes in kidney transplantation. Nevertheless, evidence supporting routine testing remains insufficient. ELISA testing for anti-AT1R and anti-ETAR antibodies was performed in a pediatric renal transplant cohort. We selected 12 pediatric recipients who had undergone protocol biopsies and antibody measurements at 6 and 24 months post-transplant. Immunohistochemistry was performed on biopsies for AT1R and ETAR as well as the adhesion molecules ICAM-1 and VCAM-1. The analysis showed that ICAM-1 and VCAM-1 expression was significantly increased, along with the presence of circulating antibodies, in patients at 24 months post-transplant compared to patients without circulating antibodies. The presence of anti-AT1R and anti-ETAR antibodies does not seem to influence the expression of their receptors in the transplanted organ. Instead, the increase in adhesion molecules may precede the development of histological damage. Therefore, enlarging the cohort and extending long-term observation would help to understand the impact of anti-AT1R and anti-ETAR antibodies after transplantation.

## 1. Introduction

Kidney transplantation is the treatment of choice for children with chronic kidney disease, providing a better quality of life and reducing morbidity and mortality compared to long-term dialysis [1]. Despite advances in surgical techniques, infection surveillance, and effective immunosuppressive therapies that have improved the short-term survival of kidney allografts in children, the long-term survival of kidney allografts has remained static to no more than 10–20 years. Events such as T-cell-mediated rejection (TCMR), the development of de novo donor-specific anti-HLA antibodies (DSA), and subsequent active antibody-mediated rejection (ABMR) are major contributors to renal allograft failure. The incidence of acute rejection is 10–20%, and early treatment with intensified immunosuppression can improve allograft survival [2,3].

Several immunological and non-immunological factors contribute to premature organ failure, with antibody-mediated rejection being the leading cause of transplant loss after the first year. Despite anti-HLA antibodies being the primary cause of antibody-mediated rejection, other non-HLA antibodies have recently been identified, which appear to play a role in humoral rejection [4]. They are directed against allogeneic molecules such as MHC class I chain-related gene A (MICA-Ab) or B (MICB), or against autoantigens like endothelin-1 type A receptor (ETAR), angiotensin II type 1 receptor (AT1R) perlecan, agrin, or vimentin, among others [3].

Anti-AT1R and anti-ETAR antibodies are the most studied; they have been detected in the sera of recipients with renal dysfunction, and they are one of the possible causes of transplant complications [5]. Nevertheless, their role is still debated: these autoantibodies are often found in healthy individuals, and not all recipients with elevated antibody levels experience renal dysfunction. Furthermore, the strong correlation between the presence of anti-AT1R and anti-ETAR autoantibodies makes it challenging to determine which one contributes more significantly to renal damage.

Anti-AT1R antibodies are the most well-known and extensively studied non-HLA antibodies. They belong to the IgG1 and IgG3 subclasses and are thought to cause endothelial damage through a complement-independent mechanism, involving the direct activation of endothelial cells and the recruitment of inflammatory cells in the graft [6].

Anti-AT1R antibodies target the angiotensin II type 1 receptor (AT1R) and recognize two distinct epitopes on the second loop of the receptor. AT1R is part of the G-protein-coupled receptor (GPCR) family, which is the largest group of cell surface receptors and represents a major target for pharmacological therapies. Under normal conditions, the activation of the receptor by angiotensin II induces vasoconstriction in smooth muscle cells and promotes vascular endothelial cell proliferation. Anti-AT1R antibodies act agonistically on the receptor, causing a prolonged allosteric effect. This results in the pathological hyperactivation of AT1R, leading to vasoconstriction that is ten times stronger than that induced by its natural ligand [7].

Recent studies have made significant efforts to understand the impact of anti-AT1R antibodies in kidney transplantation. Lefaucheur et al. examined a large cohort of kidney transplant recipients, measuring donor-specific anti-HLA and anti-AT1R antibodies either at the time of the first acute rejection episode or one year post-transplantation. Patients with AT1R antibodies exhibited a higher incidence of antibody-mediated rejection compared to those without anti-AT1R antibodies (25.0% vs. 12.9%). In addition, anti-AT1R antibodies were associated with a significantly increased risk of allograft loss [8].

In another study, anti-AT1R antibodies were investigated in a cohort of 65 pediatric renal transplant recipients, followed for 2 years post-transplantation. Anti-AT1R antibodies were detected in 58% of the children, but there was no association between anti-AT1R antibodies and donor-specific anti-HLA antibodies. The presence of anti-AT1R antibodies was linked to renal allograft loss, as well as the presence of glomerulitis or arteritis, and elevated levels of TNF-α, IL-1β, and IL-8. Despite this, anti-AT1R antibodies were not associated with rejection or hypertension. On the other hand, the production of anti-AT1R antibodies was correlated with a greater decline in graft function (eGFR) in both patients with and without rejection [9].

Anti-ETAR antibodies are IgG1 autoantibodies that bind to the second extracellular loop of ETAR. Like anti-AT1R antibodies, anti-ETAR antibodies target a member of the GPCR family: the endothelin-1 receptor. ETAR mediates several key functions, including vasoconstriction, cardiovascular remodeling, oxidative stress, mesangial cell proliferation, renal parenchymal injury, capillary leakage, cell proliferation, migration, differentiation, and renal and vascular inflammation—particularly T-cell infiltration in the kidney [6]. A high presence of anti-ETAR antibodies has been associated with rejection and functional decline, potentially leading to kidney loss, and patients with positive anti-ETAR and anti-AT1R antibodies exhibited worse organ function compared to those with only one of the antibodies [10]. A study analyzing the role of anti-ETAR antibodies in renal allograft rejection found that the presence of these antibodies was associated with poorer renal transplant function during the first 12 months post-transplantation [11].

The mechanisms of damage caused by autoantibodies have not yet been clearly defined, but anti-ETAR antibodies are supposed to induce transplant injury both through complement-dependent and complement-independent mechanisms [12]. Anti-AT1R and anti-ETAR antibodies cause the hyperactivation of their respective receptors. The prolonged activation of these receptors leads to hypertension, fibroblast migration, and the recruitment of immune cells into the kidney, resulting in increased inflammation within the organ.

The recruitment of inflammatory cells from the bloodstream is evidenced by the increased expression of adhesion molecules such as ICAM-1 and VCAM-1. ICAM-1 is the intercellular adhesion molecule-1, and its binding to integrins promotes trans-endothelial leukocyte migration and T-lymphocyte activation [13]. ICAM-1 is constitutively expressed on the surface of a wide variety of cells, particularly vascular endothelial cells and immune system cells. Its expression is upregulated by pro-inflammatory cytokines such as interleukin (IL-1), tumor necrosis factor (TNF-α), and interferon (IFN-γ). VCAM-1 is a vascular cell adhesion protein 1 that belongs to the immunoglobulin superfamily. Integrins bind to VCAM-1 only when leukocytes are activated by chemotactic agents or stimuli, often resulting from endothelial injury. The interaction between VCAM-1 and integrins is crucial for the adhesion of lymphocytes, monocytes, and eosinophils to the vascular endothelium and for promoting the migration of leukocytes towards inflamed tissues. VCAM-1 is expressed in immune response cells and is highly expressed in activated endothelial cells. Endothelial activation is induced by agents such as lipopolysaccharides, IL-1, and TNF-α [14]. Transplant rejection, initiated by leukocyte infiltration at sites of inflammation, is due to a complex interaction between the recipient’s leukocytes and the donor’s endothelium. VCAM-1 expression in the transplanted kidney is upregulated during rejection, primarily on the basolateral surface of the tubules and in the peritubular capillaries [15].

This study explored the role and involvement of anti-AT1R and anti-ETAR antibodies in pediatric renal transplantation, by evaluating the serum levels of these autoantibodies and comparing them to the expression of their target protein in protocol biopsies up to 2 years after renal transplantation.

## 2. Results

The study population comprised 167 pediatric patients, including 117 males and 50 females. Among these, 14 patients received transplants from living donors. The average age at the time of transplantation was 8 years and 8 months. All patients received basiliximab and steroids as immunosuppression induction therapy and then calcineurin inhibitor (tacrolimus or cyclosporine) and mofetil mycophenolate and steroids as maintenance therapy, according to our protocol.

For the study, a group of 12 patients (5 females and 7 males) was selected based on known levels of anti-AT1R (Angiotensin II Type 1 Receptor) and anti-ETAR (Endothelin Type A Receptor) antibodies. The source of the transplant was a living donor in three cases and a deceased donor in the remaining nine children. Primary disease leading to end-stage renal disease was mainly represented by congenital abnormalities of the kidney and urinary tract (CAKUT), such as unilateral or bilateral renal dysplasia, unilateral or bilateral renal hypoplasia, ureteral duplication, and vesicoureteral reflux. In detail, six children had CAKUT, five had a rare genetic disorder (such as autosomal recessive polycystic kidney disease, nephronophthisis, C3 glomerulopathy, and coenzyme Q10 deficiency), and one had chronic interstitial nephritis. Among the enrolled patients, six had double positivity and high levels of anti-AT1R and -ETAR antibodies (>40 U/mL) from transplantation (T0) to 24 months while the other six had negative anti-AT1R and -ETAR antibodies (<17 U/mL) from T0 to 24 months.

### 2.1. AT1R and ETAR Localization and Expression

In our cohort, AT1R and ETAR had similar localization patterns and were predominantly expressed in the tubules, with smaller amounts found in the glomeruli and vessels. Infiltrating cells, when present, expressed both receptors on their surface (Appendix A). Comparison analysis showed no differences in AT1R and ETAR tissue expression between patients with positive and negative antibodies (Figure 1). The statistical analysis confirmed no significant differences in any of the analyzed compartments between patients with high levels of anti-AT1R and anti-ETAR and patients with no autoantibodies at 6 and 24 months post-transplant.

#### ICAM-1 and VCAM-1 Localization and Expression

The analysis of our samples revealed that ICAM-1 was localized and expressed almost exclusively at the vascular level. VCAM-1, although primarily expressed at the vascular level, was also present at the tubular level and, to a lesser extent, in the glomeruli. Infiltrating cells, when present, expressed both the adhesion molecules on their surface (Figure 2). The comparison of ICAM-1 and VCAM-1 expression between patients with positive and negative anti-AT1R and anti-ETAR antibodies demonstrated a statistically significant difference both at 6 and 24 months after transplantation (Figure 3; *p*-value: 0.0022 and 0.0476).

However, in our cohort, VCAM-1 expression between patients with positive and negative anti-AT1R and anti-ETAR antibodies was statistically significant only at 24 months post-transplant (Figure 3; *p*-value: 0.0043). A Mann–Whitney U-test was performed to compare the evolution of positive and negative patients at 6 and 24 months after kidney transplantation. No statistically significant difference was found.

Among the 24 protocol biopsies evaluated, 17 were classified as Banff 1 (normal biopsy or nonspecific changes), whereas 6 as Banff 3 (borderline (suspicious) for acute TCMR) and 1 as Banff 4 (TCMR). We found no significant differences concerning histological diagnoses between the groups of pediatric patients who were positive or negative for anti-AT1R and anti-ETAR antibodies.

## 3. Discussion

One of the main causes of premature graft loss is antibody-mediated rejection (ABMR). A key factor in this type of rejection is the presence of donor-specific antibodies against human leukocyte antigens (HLAs), which are considered the most important alloantigens in transplantation [16]. In recent years, there has been a growing focus on the detection and clinical significance of non-HLA antibodies in kidney transplantation. Non-HLA antibodies, which target endothelial and epithelial proteins, may be associated with poorer outcomes in kidney transplantation. Among these, antibodies against AT1R and ETAR have emerged as the most involved. These receptors are widely expressed on the surface of cells, tissues, and organs throughout the body. In the kidney, they are particularly abundant. Their activation by natural ligands angiotensin II and endothelin-1 plays a crucial role in regulating blood flow, promoting angiogenesis, and supporting endothelial cell proliferation. The presence of antibodies against these receptors can disrupt these vital processes, potentially leading to adverse effects on graft function and overall transplant outcomes. Anti-AT1R and anti-ETAR antibody binding leads to receptor hyperactivation. This seems to result in a pro-inflammatory environment and excessive vasoconstriction. The extreme vasoconstriction can cause severe complications such as malignant hypertension [10]. Currently, there is still an ongoing debate about the clear function of non-HLA antibodies in kidney transplantation at any age. Different studies have highlighted the role of these autoantibodies in antibody-mediated rejection and premature organ failure at any age. In adults, two studies have shown how these antibodies can negatively influence graft outcomes and promote AMR [16]. In particular, a recent study reported that the presence of anti-ETAR antibodies is associated with AMR on adult kidney transplant recipients, whereas the presence of anti-AT1R antibodies is linked to an increased risk of TCMR [17]. Regarding the pediatric population, Pearl et al. showed that anti-AT1R and anti-ETAR antibodies promote renal function decline. They found that the de novo development of these antibodies can increase the risk of graft loss within two years post-transplantation [18]. However, no study has specifically correlated the presence of non-HLA antibodies with a distinct histological feature of the graft and kidney function in children.

In view of this, our study aimed to investigate whether understanding the role of non-HLA antibodies could provide useful and meaningful insights into their potential detrimental impact on the course of pediatric kidney transplantation. To unravel the involvement of anti-AT1R and anti-ETAR antibodies, we compared the antibody concentration to the protein expression on protocol biopsies. We found no significant differences between the groups of pediatric patients who were positive and negative for anti-AT1R and anti-ETAR antibodies in terms of histological diagnoses. Our results are in accordance with the report by Pizzo et al., showing that elevated levels of anti-AT1R antibodies are not associated with reduced renal function or rejection in pediatric patients [19]. However, differently from the report by Pearl et al., none of our patients showed any sign of arteritis [4]. One possible explanation of this difference might be that, in our case series, only protocol biopsies, and not for-cause biopsies were taken into consideration. Consequently, all the rejections detected in our patients were subclinical. This distinction could also explain the lack of correlation between anti-AT1R and anti-ETAR antibody positivity and organ rejection in our study.

Furthermore, unlike previous studies, our research focuses on early associations between antibody levels and biopsy lesions. Through immunohistochemical analysis, we found that AT1R was mainly expressed in mesangial cells, endothelial and smooth muscle cells of renal vessels, and infiltrating cells in the interstitium, and, to a greater extent, in tubular cells, as described previously by Sorohan et al. [10]. ETAR was highly expressed in the tubular compartment and was also present in other compartments, including the glomerulus, vessels, and infiltrating cells. A similar renal localization was reported by Banasik et al., who also demonstrated the predominant expression of ETAR in the tubular epithelium [20].

Although many studies have shown that receptor expression can be influenced by genetic and environmental factors, none have explored how high levels of anti-AT1R and anti-ETAR autoantibodies might affect protein expression. The statistical analysis of the data obtained from our semi-quantitative analysis revealed no significant differences in AT1R and ETAR expression between the anti-AT1R and anti-ETAR antibody-positive patient group and the antibody-negative group. Our study indicates that receptor expression does not increase or decrease with the presence of these autoantibodies. Further molecular biology analysis will be required to investigate more precisely the gene level regulation.

One limit of our study is the low number of patients. However, it must be pointed out that it is extremely difficult to find patients as well characterized clinically and histologically as ours in the pediatric population. Thus, although small, our study may contribute to understand the function of AT1R and ETAR and their link to their adhesion molecules ICAM-1 and VCAM-1 in pediatric kidney transplantation.

Indeed, we also successfully identified the localization of ICAM-1 and VCAM-1. These adhesion molecules are typically found on the surface of endothelial cells in blood vessels and immune system cells. ICAM-1 is predominantly localized at the vascular level, whereas VCAM-1 is more broadly distributed, also appearing on other renal cells, particularly tubular cells, and, to a lesser extent, on glomerular cells. Our statistical analysis of the semi-quantitative data revealed significant differences in the expression of adhesion molecules between the anti-AT1R and anti-ETAR antibody-positive group and the antibody-negative group. Notably, VCAM-1 expression was higher in patients with elevated levels of anti-AT1R and anti-ETAR antibodies. This difference was already observed at 6 months post-transplant, although it became statistically significant only at 24 months post-transplant (*p*-value of 0.0043).

On the contrary, ICAM-1 was differently expressed in patients with positive and negative anti-AT1R and anti-ETAR antibodies already at 6 months post-transplant (*p*-value: 0.0022), and the value remained significant until 24 months post-transplant (*p*-value: 0.0476).

In conclusion, based on the results obtained from this study focused on the pediatric kidney transplant population, it can be hypothesized that the presence of autoantibodies could increase the activation of AT1R and ETAR and this might lead to the upregulation of ICAM-1 and VCAM-1. As previously reported, the hyperactivation of AT1R and ETAR triggers a signaling cascade that stimulates the expression of transcription factors such as AP-1 and NF-kB. These factors are responsible for the increased gene expression of pro-inflammatory mediators like MCP-1 and RANTES, which are crucial mediators of the pro-inflammatory immune response, promoting the production of cytokines such as IL-1, TNF-α, and IFN-γ. These cytokines play a key role in inducing microvascular inflammation in the glomerulus and peritubular capillaries, contributing to endothelial damage [14,18]. The presence of pro-inflammatory cytokines and endothelial damage leads to an increase expression of adhesion molecules ICAM-1 and VCAM-1, which are essential for the trans-endothelial migration of immune cells. Based on the results of this study, it is plausible to assume that this mechanism may begin within the first two years post-transplantation, even in patients with stable renal function.

## 4. Materials and Methods

### 4.1. Population

A retrospective study was conducted on children who underwent kidney transplantation at Padua Pediatric Center between January 2011 and June 2022. The mean age of the patients at the time of transplantation was 10 years (2–18 years old).

According to our practice, a protocol biopsy was performed 6, 12, and 24 months after transplantation. In addition, a serum sample was stored at the same follow up points. All samples were collected after informed consent from the parents. Biopsies were obtained under ultrasound guidance using an automated gun with a 16 g needle, fixed in 8% neutral non-buffered formalin, and then embedded in paraffin. Then, 4 μm paraffin sections were prepared and stained with hematoxylin and eosin, Periodic Acid–Schiff (PAS), Silver, and Masson trichrome (TRI) histological stains, and graded by an inhouse renal pathologist. The immunohistochemical staining of the complement C4d fraction deposits on peritubular capillary was performed on the paraffined sections using a peroxidase-conjugated anti-C4d monoclonal antibody (DB BIOTECH, Košice, Slovak Republic) and Peroxidase Diaminobenzidine (DAB) as chromogenic substrate (Dako, Agilent, Santa Clara, USA). Biopsy specimens were graded according to the Banff 2017 classification [21].

Patients for whom the formalin-fixed, paraffin-embedded tissue was available from protocol renal biopsies at 6 and 24 months after transplantation and who had a paired serum sample stored at the same follow-up time were considered for the study. Data regarding the immunosuppression schedule (induction and maintenance therapy), the source of the transplant, and ESRD-causing disorders were revised for all selected patients.

### 4.2. ELISA Assay

Enzyme-Linked Immunosorbent Assays (ELISAs) were conducted to detect anti-AT1R and anti-ETAR antibodies in the sera of pediatric transplant patients. The study exploited enzyme immunoassays (EIAs) for the quantitative determination of anti-Angiotensin II Type 1 Receptor (AT1R) antibodies and anti-Endothelin Receptor A (ETAR) antibodies (CellTrend GmbH, Luckenwalde, Germany), following the manufacturer’s instructions. Serum samples were diluted 1:100 and loaded in duplicate into the assay wells. A positive control, a negative control, standard curves with antiAT1R and anti ETAR-antibodies (calibration from 2.5 to 40U/mL) were included in each plate. The results were evaluated according to the criteria outlined in Table 1.

Based on the obtained values, 6 out of the 12 patients exhibited high levels both of anti-AT1R and anti-ETAR antibodies (>40 U/mL) throughout the follow-up period and were classified as “positive”, whereas 6 patients had consistently low antibody levels (<17 U/mL) and were classified as “negative”. All selected patients demonstrated either double positivity or double negativity for anti-AT1R and anti-ETAR antibodies. All 12 patients were negative for HLA-DSA antibodies.

### 4.3. Immunohistochemistry

To check whether high levels of autoantibodies could influence either the expression of their receptors or the transduction of their signal, a semi-quantitative immunohistochemical analysis was conducted for AT1R and ETAR and two adhesion molecules ICAM and VCAM expressed downstream of the signal. Formalin-fixed paraffin-embedded tissue (FFPE) sections (4 µm) of renal protocol biopsies were considered for the analysis. All primary antibodies (GeneTex San Antonio, CA, USA; listed in Table 2) were diluted at 1:200. Antigen retrieval was performed using 10 mM Na-Citrate buffer (pH 6.0). Horseradish peroxidase (HRP)-conjugated was used as secondary antibody (EnVision^®^+ System-HRP Labelled Polymer anti-rabbit, Dako, Agilent, Santa Clara, CA, USA) and visualized by 3,3′-diaminobenzidine (Dako, Agilent, Santa Clara, CA, USA). Nuclei were stained with hematoxylin (DIAPATH Spa, Bergamo, Italy).

The slides prepared by immunohistochemistry were analyzed using a semi-quantitative microscopic evaluation. Four tissue compartments were assessed: the glomerulus, tubules, vessels, and interstitium. The scoring was carried out using a four-point scale based on the IHC intensity of the staining: negative (0), mild positivity (1+), moderate positivity (2+), and strong positivity (3+). For each slide, 10 random fields were evaluated at 100× magnification. The total score was assigned to each slide, which was then divided by the four compartments to obtain a final average value based on the slide’s overall color intensity. Observations were conducted using a Leica DMLB 30W optical microscope connected to a Leica DFC450C camera (Leica-microsystems, Wetzlar, Germania). Image analysis was performed with the Leica Application Suite V4 software (Leica-microsystems, Wetzlar, Germania).

### 4.4. Statistical Analysis

Statistical analysis was performed using GraphPad Prism software version 9.4.1. The Shapiro–Wilk test was used to assess the normality of the data distribution. As the data were not normally distributed, a non-parametric two-tailed Mann–Whitney U test for unpaired samples was conducted to analyze the differences in the mean values between the positive and negative sample groups, with a significance level of α = 0.05.

## 5. Conclusions

In the protocol biopsies of our patients, we observed an increase in the expression of adhesion molecules ICAM and VCAM in the presence of high antibody titers of non-HLA anti-AT1R and anti-ETAR antibodies. These antibodies permanently bind to AT1 and ETA receptors, causing hyperactivation, and could determine a pro-inflammatory environment characterized by the production of pro-inflammatory cytokines. This environment could contribute to microvascular inflammation and increase the expression of intercellular adhesion molecule-1 (ICAM-1) and vascular cell adhesion molecule-1 (VCAM-1). The increase in adhesion molecules is usually associated with the increased migration of immune cells that, once out of the circulatory system, migrate towards the damaged organ. In conclusion, anti-AT1R and anti-ETAR antibodies in our cohort could be considered potential risk factors for allograft transplantation. However, the analysis of a larger cohort of patients and longer follow-up are needed to improve our understanding of the role of these autoantibodies and to support the inclusion of anti-AT1R and anti-ETAR antibody tests in the monitoring of post-transplant patients, in addition to standard anti-HLA antibody tests.

## Figures and Tables

**Figure 1 ijms-25-11817-f001:**
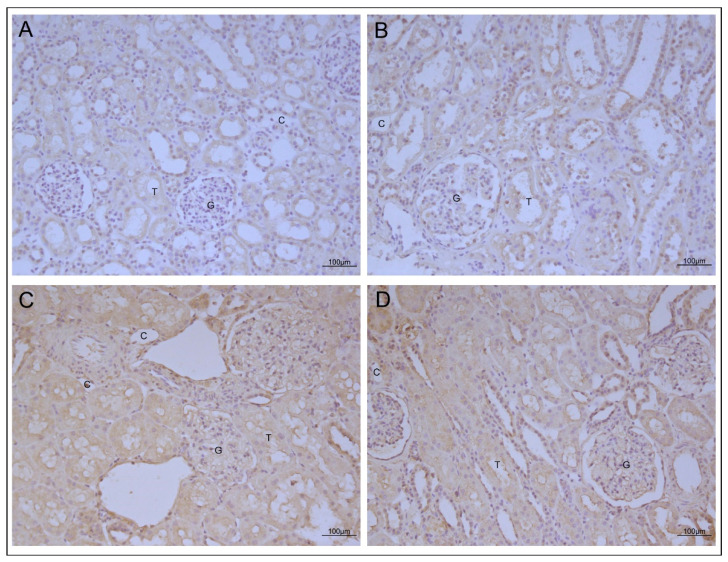
Immunohistochemical representative images of the AT1R and ETAR tissue expression. (**A**) shows AT1R staining in a seropositive patient. (**B**) shows AT1R staining in a seronegative patient. (**C**) shows ETAR staining in a seropositive patient. (**D**) shows ETAR staining in a seronegative patient (figure magnification 20×; legend: G = glomerulus; C = capillary; T = tubulous).

**Figure 2 ijms-25-11817-f002:**
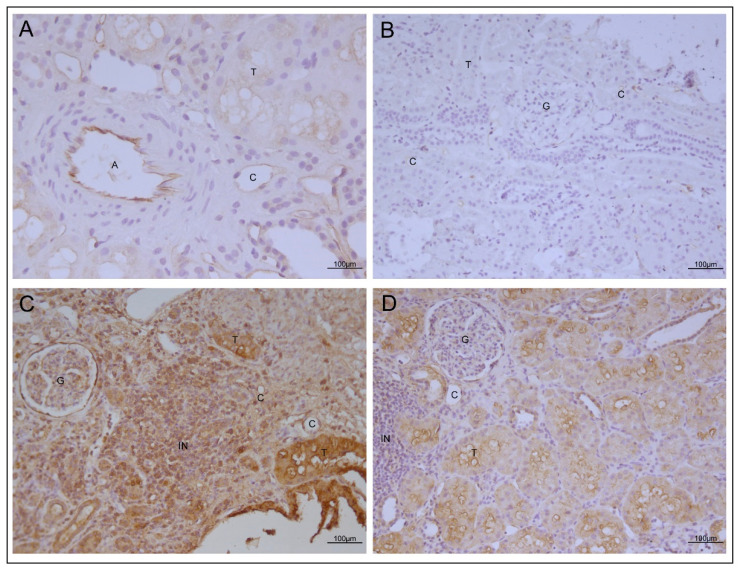
Immunohistochemical representative images of the ICAM-1 and VCAM-1 tissue expression. (**A**) shows ICAM-1 staining in a seropositive patient. (**B**) shows ICAM-1 staining in a seronegative patient. (**C**) shows VCAM-1 staining in a seropositive patient. (**D**) shows VCAM-1 staining in a seronegative patient. (Figure (**A**): magnification 40×; figures (**B**–**D**): magnification 20×; legend: A = Artery; G = glomerulus; C = capillary; IN = inflammatory infiltrate; T = tubulous).

**Figure 3 ijms-25-11817-f003:**
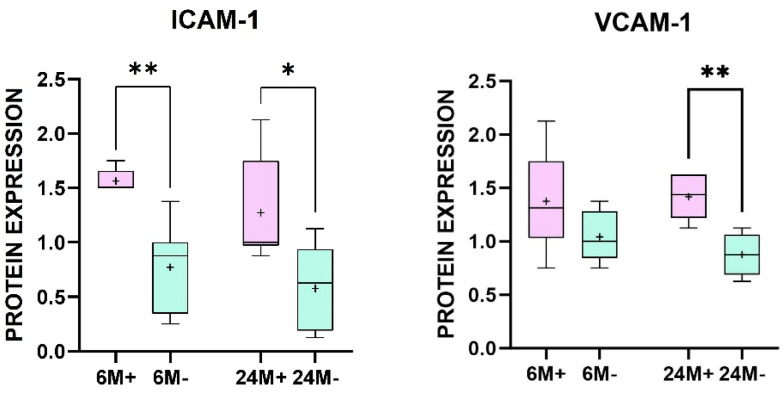
Boxplot graphs representing the differences in ICAM1 and VCAM1 expression 6 and 24 months after kidney transplantation in patients with serum levels > 40 U/mL of anti-AT1R and ETAR antibodies (pink) or negative for anti-AT1R and ETAR antibodies (green). The boxplots have IQ1 and IQ3 as lower and upper extremes, respectively. The median divides the box into two parts and it represents IQ2. The whiskers are obtained by connecting IQ1 to the minimum and IQ3 to the maximum. The + (in the boxes) represents the meaning (legokend: * *p* < 0.05; ** *p* > 0.005; M: months after transplant; signs + or − representing patients positive (+) or negative (−) for serum levels of anti-AT1R and anti-ETAR antibodies).

**Table 1 ijms-25-11817-t001:** Test interpretation cut-off and division of subgroups based on the antibody concentration levels.

Antibodies Concentration	Subgroup
<17 U/mL	negative
17–40 U/mL	positive
>40 U/mL	highly positive

**Table 2 ijms-25-11817-t002:** Antibodies used in immunohistochemistry assay.

Company	Cat. No	Antibody	Host	Reactivity	Clonality	Isotype	MW (kDa)
GeneTex	GTX31793	AGT1R	Rabbit	Human, Mouse, Rat	Polyclonal	IgG	41
GeneTex	GTX116034	Endothelin A Receptor (C3)	Rabbit	Human, Mouse, Rat, Chicken	Polyclonal	IgG	49
GeneTex	GTX100450	ICAM1/CD54	Rabbit	Human, Mouse	Polyclonal	IgG	58
GeneTex	GTX110684	VCAM1/CD106	Rabbit	Human	Polyclonal	IgG	81

## Data Availability

The data presented in this study are available on request from the corresponding author. The data are not publicly available due to privacy restrictions.

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
