# Peer review of "Non-HLA Autoantibodies Against Angiotensin II Receptor 1 (AT1R) and Endothelin A Receptor (ETAR) in Pediatric Kidney Transplantation"

_ijms, 2024, doi:10.3390/ijms252111817_

Round 1
Reviewer 1 Report
Comments and Suggestions for Authors
Comments
In the case report “Non-HLA autoantibodies against angiotensin II receptor 1 (AT1R) and endothelin A receptor (ETAR) in pediatric kidney transplantation” Antoniello et al. present evidence that anti-AT1R and anti-ETAR antibodies are associated with increased ICAM-1 and VCAM-1 expression in the graft following kidney transplantation.
Even though, the results are interesting they remain indicative without the use of quantitative analyses (e.g. qPCR, RNAseq) in a larger pool of patients.
However, the objectives and results are thoroughly presented. So, the authors are requested to add details on the following points:
1. References showing that non-HLA Abs, e.g. anti-AT1R or anti-ETAR, cause ABMR (lines 57-59)
2. What is the type of AT1R and ETAR receptors? Also what pathway is activated by targeting these receptors and what physiologic responses are produced from the activation of the receptors by the auto-antibodies?
3. Is there evidence of cross-talk between these receptors?
4. Are the levels of these Abs age-dependent? If so, a note should be included that presented results may not be true in older patients
5. It would be interesting to correlate the levels of these Abs in the presence of HLA in patient sera.

Author Response
Thanks for the suggestions. We have inserted the missing reference and responded as indicated below.
Revisor 1:
However, the objectives and results are thoroughly presented. So, the authors are requested to add details on the following points:
- References showing that non-HLA Abs, e.g. anti-AT1R or anti-ETAR, cause ABMR (lines 57-59)
A: We added the missing reference and a sentence to specify. (line 61).
- What is the type of AT1R and ETAR receptors? Also what pathway is activated by targeting these receptors and what physiologic responses are produced from the activation of the receptors by the auto-antibodies?
A: They are both G protein-coupled receptors (missing information added). We added also the physiological response for AT1R (lines 75-85). For ETAR we added a sentence to specified (line 104) but the physiologic response was present (lines 105-109).
- Is there evidence of crosstalk between these receptors?
A: There is no reported evidence of a crosstalk between these receptors.
- Are the levels of these Abs age-dependent? If so, a note should be included that presented results may not be true in older patients.
A: The youngest patient is 2 years old, and the oldest one is 18 years old (We added this information in MM paragraph). We didn’t observe an age-dependent correlation in the 12 patients analysed in the study.
- It would be interesting to correlate the levels of these Abs in the presence of HLA in patient sera.
A: All 12 patients analysed in IHC were negative for HLA-DSA antibodies (added in line 364).

Reviewer 2 Report
Comments and Suggestions for Authors
The correlational study by Antoniello et al. describes differences in ICAM-1 and VICAM-1 protein levels in biopsy tissue samples collected from pediatric kidney transplant recipients that either have or not alloantibodies to AT1R and ATAR. The paper is very well written and presented data are clear. I only have minor comments, which should be addressed to make the paper suitable for publication:
11. Please mention that other non-HLA antibodies exist beyond those against AT1R and ETAR.
22. There are few typos:
a. Line 28: please remove “and”.
b. Line 31: were statistically significant together as compared to …?
c. Please put reference at line 70.
d. Line 144: correct the sign “>” to “<”.
e. There are instances in the text where ICAM or VCAM are written without 1 (Fig 3), lines 330-331, ….
f. Line 212 and 131 remove period before reference [14] and [18].
g. Line 229 add “s” to patient.
33. Figures:
a. Figure legends need to be rewritten as they are very vague/confusing:
i. Figure 1 “A: AT1R patient…” refers to AT1R staining of seropositive patient, and “B: AT1R patient…” refers to AT1R staining of seronegative patient.
b. Figure 2A seems to be of greater magnification.
c. Please provide scale bars to Figure 1 and 2.
d. Please provide arrows, stars that can help identify explicitly in IHC images the structures mentioned in the paper (tubules, glomeruli, vessels).
e. Figure 3 please explain what the boxplot graphs are depicting: median, IQR,. What is the meaning of the “+” sign. Please decide how the statistical difference is shown (asterisk or numerical values).
44. M&M:
a. Specify the name of the center on line 292 and remove “our”.
b. It is unclear if the antibody concentration levels are calculated cumulatively (anti-AT1R + ETAR >40U/ml) or individually (AT1R>40 U/ml ETAR>40U/ml). Please clarify and provide the detection limit of the EIA kits.
c. In Table 2 provide specific category numbers for each product.
d. Lines 254-256: the sentence starting with “However, …” is not necessary.
55. General considerations:
a. For Figure 3 did the authors consider performing statistical analysis of not one parameter (POS vs NEG) but two parameters (POS vs NEG and 6M vs 24 M) by 2-way ANOVA to see if the protein levels change with time? Although the sample size (n=6) is small, ICAM-1 levels appear to decrease in both groups, while VCAM-1 levels decrease in the NEG group but remain stable in the POS group.
b. It would have been good to provide a staining from healthy kidney (ideally age-matched controls) and quantify AT1R, ETAR, ICAM-1 and VCAM-1 staining levels.
Author Response
Thank you for the suggestions. We added and corrected in response to your observations.
- Please mention that other non-HLA antibodies exist beyond those against AT1R and ETAR.
A: Mentioned in lines:59-63.
- There are few typos:
- Line 28: please remove “and”.
A: Corrected.
- Line 31: were statistically significant together as compared to …?
A: We modified the sentence in “However, ICAM-1 and VCAM-1 expression was statistically significantly increased, along with the presence of circulating antibodies, in patients with circulating antibodies at 24 months post-transplant compared to patients without circulating antibodies.”
- Please put reference at line 70.
A: Added ref 5
- Line 144: correct the sign “>” to “<”.
A: Corrected-
- There are instances in the text where ICAM or VCAM are written without 1 (Fig 3), lines 330-331,
A: Corrected.
- Line 212 and 131 remove period before reference [14] and [18].
A: Corrected.
- Line 229 add “s” to patient.
A: Corrected.
- Figures:
Figure legends need to be rewritten as they are very vague/confusing:
- Figure 1 “A: AT1R patient…” refers to AT1R staining of seropositive patient, and “B: AT1R patient…” refers to AT1R staining of seronegative patient.
A: Rewriten.
- Figure 2A seems to be of greater magnification.
A: Yes, we added the information.
- Please provide scale bars to Figure 1 and 2.
A: Provided.
- Please provide arrows, stars that can help identify explicitly in IHC images the structures mentioned in the paper (tubules, glomeruli, vessels).
A: added in the images.
- Figure 3 please explain what the boxplot graphs are depicting: median, IQR,. What is the meaning of the “+” sign. Please decide how the statistical difference is shown (asterisk or numerical values).
A: Explained in the legend.
- M&M:
- Specify the name of the center on line 292 and remove “our”.
A: Specified.
- It is unclear if the antibody concentration levels are calculated cumulatively (anti-AT1R + ETAR >40U/ml) or individually (AT1R>40 U/ml ETAR>40U/ml). Please clarify and provide the detection limit of the EIA kits. The levels are not cumulative (added “both” at line 358).
A: We consider individual dosages of anti-AT1R and anti-ETAR and “All selected patients demonstrated either double positivity or double negativity for anti-AT1R and anti-ETAR antibodies”(line 351).
- In Table 2 provide specific category numbers for each product.
A: inserted the cat number.
- Lines 254-256: the sentence starting with “However, …” is not necessary.
A: Deleted.
- General considerations:
- For Figure 3 did the authors consider performing statistical analysis of not one parameter (POS vs NEG) but two parameters (POS vs NEG and 6M vs 24 M) by 2-way ANOVA to see if the protein levels change with time? Although the sample size (n=6) is small, ICAM-1 levels appear to decrease in both groups, while VCAM-1 levels decrease in the NEG group but remain stable in the POS group.
A: The ANOVA test could not be performed because the data were not normally distributed. However, Mann Whitney's U-test was performed to compare the evolution of positive and negative patients at 6 and 24 months after kidney transplantation. No statistically significant difference was found. We added this part to the results (line 181).
- It would have been good to provide a staining from healthy kidney (ideally age-matched controls) and quantify AT1R, ETAR, ICAM-1 and VCAM-1 staining levels.
A: Unfortunately, using human biopsies of healthy kidneys is impossible. In the literature, biopsies of a pole not damaged by a nephrectomy performed on a patient with cancer are sometimes used, but their use is very controversial. We tried to use the renal biopsy performed at time 0 of the transplant. However, even in this case, the ischemia and reperfusion time often reaches 6-8 hours and, in the kidney, significant acute tubular damage is evidenced, leading the tissue to have a very high and confused background.
